

# Enrichment of polycyclic aromatic hydrocarbon metabolizing microorganisms on the oral mucosa of tobacco users

Lin Tao[1,*], M Paul Chiarelli[2,*], Sylvia Pavlova[1], Antonia Kolokythas[3], Joel Schwartz[4], James DeFrancesco[5], Benjamin Salameh[4], Stefan J. Green[6] and Guy Adami[4]

[1] Department of Oral Biology, College of Dentistry, University of Illinois Chicago, Chicago, IL, United States of America
[2] Department of Chemistry and Biochemistry, Loyola University of Chicago, Chicago, IL, United States of America
[3] Department of Oral and Maxillofacial Surgery, Eastman Institute for Oral Health, University of Rochester, Rochester, NY, United States of America
[4] Oral Medicine and Diagnostic Sciences, University of Illinois Chicago, Chicago, IL, United States of America
[5] Forensic Science Program | Department of Criminal Justice, Loyola University of Chicago, Chicago, IL, United States of America
[6] DNA Sequencing Core, Research Resources Center, University of Illinois Chicago, Chicago, IL, United States of America
[*] These authors contributed equally to this work.

Corresponding author
Guy Adami, gadami@uic.edu

## ABSTRACT

Certain soil microbes resist and metabolize polycyclic aromatic hydrocarbons (PAHs). The same is true for a subset of skin microbes. In the human mouth, oral microbes have the potential to oxidize tobacco PAHs, thereby increasing these chemicals' ability to cause cancer of adjacent epithelium. We hypothesized that we could identify, in smokers, the oral mucosal microbes that can metabolize PAH. We isolated bacteria and fungi that survived long-term in minimal media with PAHs as the sole carbon source, under aerobic conditions, from the oral mucosa in 17 of 26 smokers and two of 14 nonsmokers. Of bacteria genera that survived harsh PAH exposure *in vitro*, most were found at trace levels, except for *Staphylococcus*, *Actinomyces*, and *Kingella*, which were more abundant. Two PAH-resistant strains of *Candida albicans (C. albicans)* were isolated from smokers. *C. albicans* was a prime candidate to contribute to carcinogenesis in tobacco users as it is found orally at high levels in tobacco users on the mucosa, and some *Candida* species can metabolize PAHs. However, when *C. albicans* isolates were tested for metabolism of two model PAH substrates, pyrene and phenanthrene, they were not capable, suggesting they cannot metabolize PAH under the conditions used. In conclusion, evidence for large scale microbial degradation of tobacco PAHs under aerobic conditions on the oral mucosa remains lacking, though nonabundant PAH metabolizers are certainly present.

## INTRODUCTION

Years ago, it was recognized that exposure to tobacco smoke has both short- and long-term effects on the oral cavity. These effects have been best described for the periodontia where, in long-time smokers, there is an increase in inflammation at the tooth mucosal border that precedes a loss in tissue (*Brook, 2011*). There are also changes in bacteria that exist in the sulci that surround the teeth. Some of the changes are directly related to tobacco smoke exposure (*Kumar et al., 2011*; *Van Winkelhoff et al., 2001*; *Zambon et al., 1996*). Similarly, the remaining oral mucosa lining the oral cavity also shows changes with tobacco smoke exposure (*Taybos, 2003*). Initial long-term mucosal changes include hypertrophy and hyperkeratosis. These effects can be followed by more serious changes including dysplasia, which can precede malignancy. Certain sites thought to be most heavily exposed to cigarette smoke include those frequently seen with oral squamous cell carcinoma (OSCC), such as the tongue, floor of mouth (FOM) and, to a lesser degree, gingiva. These surfaces also are coated with bacteria, though the exact makeup of the population depends on the site (*Segata et al., 2012*; *Thomas et al., 2014*; *Yu et al., 2017*). Other microorganisms, such as the yeast *C. albicans*, can also become a large part of the oral microbiome in certain individuals. The microbes in the saliva provide a window to the oral microbiome that exists on the oral surfaces, including the teeth, the periodontal sulci, and the remaining oral mucosa (*Wu et al., 2016*). Portions of this text were previously published as part of a preprint (https://doi.org/10.21203/rs.3.rs-929320/v1).

There is extensive research on how combusted tobacco products provide a stream of reactive chemicals to the oral tissue (*Ding et al., 2005*; *Rodgman, Smith & Perfetti, 2000*). These include PAHs and nitrosamines, which have received the most attention because they are pro-carcinogens. Given that studies have revealed the existence of environmental microorganisms that have the ability to metabolize PAHs (*Ghosal et al., 2016*; *Habe & Omori, 2003*; *Hadibarata et al., 2017*; *Hennessee & Li, 2016*; *Hesham et al., 2009*; *Husain, 2008*; *MacGillivray & Shiaris, 1993*; *Mallick, Chakraborty & Dutta, 2011*; *Mbachu, Chukwura & Mbachu, 2016*; *Sowada et al., 2014*; *Sutherland, 1992*) and that bacteria with similar qualities exist on human skin (*Sowada et al., 2017*; *Sowada et al., 2014*) we looked for evidence of PAH metabolizing microbes on the oral mucosa. We used the same method that is used to detect PAH metabolizing microbes from environmental sites such as petroleum waste sites. Oral samples were incubated long term with PAHs as the sole carbon source to select for PAH metabolizers (*Hennessee & Li, 2016*; *Husain, 2008*; *Juhasz, Stanley & Britz, 2000*; *Sowada et al., 2014*).

In the current work, microorganisms were harvested from the oral mucosa surfaces and then exposed over weeks *in vitro* to a cocktail of PAHs with no other carbon source to determine if smokers harbored more microbes which survived better in the presence of tobacco smoke PAHs than nonsmokers. A second version of this assay identified microbes that thrived with PAH as the sole carbon source, making these taxa, *Micrococcus luteus and Kingella,* more likely to be PAH metabolizers. Most of the PAH selected taxa were found, at trace levels on the tongue and gingival mucosa even in tobacco users. *C. albicans* was found to be resistant to long term incubation with PAH. As a yeast known to be at relatively high

levels in the oral cavity of tobacco users, an examination the ability of *C. albicans* isolated from smokers to metabolize model PAHs was undertaken in order to examine a role for this species in the conversion of these PAHs to more reactive carcinogens (*Darwazeh, Al-Dwairi & Al-Zwairi, 2010*; *Sheth et al., 2016*).

## MATERIALS & METHODS

### Donors

Donors were patients of Dental Clinics at the University of Illinois College of Dentistry or the University of Rochester. All donors provided written informed consent to participate in accordance with guidelines of the Office for the Protection of Research Subjects of the University of Illinois Chicago, with formal approval of the study protocol, 2012-1030, by the Institutional Review Board 1 of the University of Illinois Chicago and RSRB case number 00006443 by the Institutional Review Board at the University of Rochester. This study was done in full accordance with the principles of the Declaration of Helsinki. Adult patients were included as they appeared in the general dentistry clinics. Included were tobacco users who smoked at least 10 cigarettes per day for one year. Nonsmokers had not smoked in the last year. Exclusion criteria: acute periodontitis, active oral infection, usage of antibiotics in the last 30 days, and usage of mouth rinse that day. In the first study, age or gender of the patients was not recorded. In the second study, patients raged in age from 38 to 70 years old, six out of 12 smokers and were female and five out of eight nonsmokers were female. Samples were taken only from sites that appeared normal in appearance in the clinic.

### Mucosal sample collection and selection

For the PAH selection procedure, swab samples were collected from the lateral border of the tongue, buccal, attached gingiva, and oral pharynx, with a single swab over 10 to 15 s, placed in PBS, vortexed and centrifuged at 5,000 xg 5′ washed with PBS, then placed in 3 mL Bushnell-Haas Broth (BHB) without glycerol. A 1 mL culture was used to inoculate 9 mL BHB with 10 µg/mL each of benzo[a]pyrene, chrysene, fluoranthrene, naphthalene, phenanthrene, and pyrene. The culture was incubated capped with aeration for 3 weeks at 37 °C in 100 ml bottles. One hundred microliters were collected and plated on BHI agar plate and incubated for 48 h at 37 °C in aerobic conditions. Samples used directly for 16S rRNA gene analysis were from gingiva or tongue and were immediately frozen in Tris-EDTA.

The second version of the PAH selection procedure was done as above with a number of differences. Patients were exclusively from the University of Rochester Dental Clinics, swabbing time was lengthened to 30 s to increase starting number of microbes and was from the lateral border of the tongue. Resuspended material (0.2 ml), after twice washing with phosphate buffered saline (PBS) was inoculated in two mL BHB with and without PAH and incubated at 37 °C for 5 weeks shaking in 15 mL tubes. Samples were purified as below and subjected to 16S rRNA gene analysis with next-generation sequencing, NGS, to identify taxa.

## PAH degradation by microbes in liquid culture

*C. albicans* was grown overnight in Yeast Extract Peptone Dextrose (YPD broth at 30 °C with aeration). The positive control, *Mycobacterium rutilum* -was prepared in BHI medium and incubated overnight at 30 °C with aeration (*Hennessee & Li, 2016*). The overnight cultures of *C. albicans* and *Mycobacterium rutilum* were centrifuged, washed three times with BHB medium, and resuspended in BHB. Each culture was divided into nine large glass test tubes, each containing 10 mL culture in BHB medium. Pyrene or phenanthrene was added (triplicates) to a final concentration of 20 μg/mL. BHB medium with microbial cells without PAH (triplicates) and BHB medium only served as controls. Tubes were incubated with aeration for 14 days in the dark at 30 °C.

## Identification of pyrene and phenanthrene metabolites

After the PAHs were incubated with the microbes, they were stored unopened at −80 °C until analyzed by GC/MS. Once the samples thawed, the cultures were spiked with 20 μg of phenanthrene d-10 or pyrene d-10 prior to extraction to allow estimation of the recovery of the unreacted pyrene and phenanthrene. After spiking with the deuterated standard, the samples were vortexed and stored overnight in a refrigerator. Each of the conical glass vials containing samples were acidified to a pH of 2 and extracted three times with three mL of MTBE (methyl-tert butyl ether, a total of nine mL). The MTBE/aqueous mixtures were centrifuged at 900 g for 10 min to separate the aqueous and MTBE layers. The MTBE extracts were combined and allowed to evaporate overnight in a standard hood and the following morning the remaining MTBE was evaporated under nitrogen. The test tubes containing the residual solid were rinsed (vortexed) with two 400 μl aliquots of MTBE. The MTBE was transferred to an autosampler vial (with a borosilicate glass insert) and evaporated to dryness in a tabletop centrifuge. This concentration and transfer step was repeated twice. Analytical procedures were designed to analyze oxidized (hydroxylated) phenanthrene and pyrene metabolites.

The samples were then derivatized (silylated) with *N,O*-Bis(trimethylsilyl)-trifluoroacetamide (BSTFA) with 1% trimethylsilyl chloride (TMSC) to convert the hydroxyl groups to trimethyl silyl groups. Extracts in the vials mentioned above were (10 μl BSTFA/TMCS in 200 μl $CHCl_3$). These samples were diluted 1/20 and 1/50 in chloroform and analyzed by GC/MS for the purpose of quantifying the unreacted PAHs. The most concentrated, undiluted samples were run as well for the purpose of identifying metabolites.

Derivatized extracts were analyzed using an Agilent 5977 Mass selective detector interfaced to an Agilent 7800 GC. A 30-meter Agilent DB5-MS column was used for the separation (Agilent, Santa Clara, CA, USA). The mobile phase was helium at a flow rate of one mL/min. The injector temperature was 240 °C. A 20-min GC gradient was used for the separation with a helium flow rate of one mL/min. The initial temperature was held at 60 °C for two minutes and then increased to 320 °C at a rate of 20°/minute and held at 320 °C for 10 min. Full scans were acquired for all analyses.

## DNA extraction

Genomic DNA was extracted from swabs using the MasterPure Gram Positive DNA Purification Kit (Epicentre, Madison, WI, USA) according to manufacturer's instructions providing both gram-positive and -negative bacteria DNA. As instructed, lysozyme was used to digest cell walls and proteinase K was used to reduce protein levels, but no ribonuclease digestion was performed.

Samples harvested from *in vitro* incubations, at least two colonies on BHI agar plates, were collected and sample DNA was extracted using alkaline lysis and Express Matrix purification (MP Biomedical, Santa Ana, CA, USA). Identification of clones was done after PCR amplification using consensus 16S rRNA primers.

16S-27F 5′-AGAGTTTGATCCTGGCTCAG-3′ and16S-1329R 5′-TCTACGCATTCCA CCGCTAC-3′ or, in the case of *C. albicans,* 28S rRNA fungal specific primers NL-1 5′-GCATATCAATAAGCGGAGGAAAAG-3′ and NL-4 5′-GGTCCGTGTTTCAAGACGG-3′ used for Sanger Sequencing (*Kurtzman & Robnett, 1997*). For samples harvested from *in vitro* incubations for the second PAH culture selection protocol, bacteria were harvested directly from culture media after centrifugation and resuspension using mechanical lysis and the Zymo Quick-DNA Fungal/Bacterial Kits as described (Zymo Research, Irvine, CA, USA) (*Adami, Ang & Kim, 2021*).

## Characterization of microbial community structure

Microbial community structure was characterized using high-throughput sequencing of PCR amplicons generated from the V1-V3 variable regions of bacterial 16S rRNA genes. Briefly, the widely used primer sets 27F/534R, targeting the V1-V3 variable region of the 16S rRNA gene of bacteria, were used for amplification as done earlier (*Adami, Ang & Kim, 2021*; *Yildirim et al., 2010*). Alternatively, the V4 variable region of the 16S rRNA gene of bacteria was amplified and sequenced using the 515F/806R primer set (*Caporaso et al., 2011*). A two-stage PCR or targeted amplicon sequencing (TAS) approach was performed to generate amplicon libraries, as described previously (*Bybee et al., 2011*; *Green, Venkatramanan & Naqib, 2015*).

Sequencing was performed on an Illumina MiSeq sequencer using standard V3 chemistry with paired-end, 300 base reads or on an Illumina MiniSeq sequencer with high output reagent and 150 cycles. Fluidigm sequencing primers, targeting the CS1 and CS2 linker regions, were used to initiate sequencing. Demultiplexing of reads was performed on instrument. Library preparation was performed at the DNA Services Facility at the University of Illinois, Chicago.

## Bioinformatics analysis

Raw paired-end FASTQ files were merged using the Paired-End reAd merger (PEAR) algorithm (*Zhang et al., 2014*). Merged data were then quality trimmed (Q20), and sequences shorter than 450 bases were removed. The remaining sequences were exported as FASTA and processed through the software package QIIME (v1.8.0) (*Caporaso et al., 2010*). Sequences were screened for chimeras using the USEARCH61 algorithm and putative chimeric sequences were removed from the data set (*Edgar, 2010*). Chimera-free

samples were then pooled, and clustered into operational taxonomic units (OTU) at 97% similarity using USEARCH assigned to taxonomic levels from phylum to species.

### Statistical analysis

BIOMs were used to identify taxa that were significantly differentially abundant between a priori defined groups. Differences in microbiota taxonomic abundance between the groups were tested using Welch's $t$-test using the software package STAMP (*Parks et al., 2014*). Significance was set at $P < 0.05$. to examine. To compare profiles of *in vitro* PAH selected taxa from tongue swabs, *versus* that from selection without PAH, both DeSeq2 and EdgeR were used after elimination of taxa with less than 100 sequence reads in a sample.

G*Power was used for sample size calculation and effect sizes were determined as described (*Chinn, 2000*; *Faul et al., 2007*).

## RESULTS

### Survival of sampled oral mucosal bacteria after long-term PAH exposure

Studies of environmental and skin microorganisms have revealed bacteria that survive in high levels of PAHs can use these chemicals as a nutrient source (*Husain, 2008*; *Sowada et al., 2017*). We took advantage of that knowledge to investigate behavioral differences of the oral bacteria associated with long-term exposure to tobacco smoke PAHs *versus* minimal exposure. First, we took swab samples from the oral mucosa of six nonsmokers and 14 smokers. The washed microorganisms were than inoculated into minimal broth and allowed to incubate for 3 weeks in the presence of a cocktail of PAHs as the only carbon source. At the end of the incubation time, the cells were plated on rich medium agar plates (BHI) for 48 h to allow the detection of surviving bacteria. With 20 subjects at a power of 80% and significance level of 5% one would be able to detect a difference presuming 70% of smokers have taxa that survive the PAH assay and 10% of nonsmokers do (*Faul et al., 2007*).

We found that PAH selection indeed allowed the survival of several bacteria that were identified by 16S rRNA gene Sanger sequencing. For smokers' samples, eleven out of fourteen produced colonies under PAH selection; these were identified as *Acinetobacter junii, Acinetobacter baumannii, Agrobactrium tumerfaciens, Actinomyces, Bacillus pumillus, Bacillus subtilis, Kocuria rhizophila, Rhodococcus, Staphylococcus epidermidis*, and *Staphylococcus*. In addition, multiple isolates from smokers revealed the presence of the yeast *C. albicans*, which was identified by 28S rRNA gene sequencing. In nonsmoker samples, one out of six grew colonies with PAH selection, which was identified as *Rhodococcus*. Smokers showed higher numbers of these microbes that could survive these harsh selective conditions where PAH was the only carbon source (Fisher's Exact Test, $p < 0.018$). The odds ratio for being positive in the PAH selection test if the subject smoked *versus* did not smoke was 18.3 with a Cohen's effect size of 1.61 (*Chinn, 2000*).

### Differences in oral mucosal bacteria with long-term tobacco usage

Having identified oral bacteria on the oral mucosa of smokers as resistant to PAH exposure, the question remained whether these bacteria comprised a significant level of the mucosal

**Table 1  Levels of PAH-selected bacteria on gingiva.**

| Genus | Percentage level[a] Never smoker | Percentage level Smoker | Detected[b] | Probability[c] |
|---|---|---|---|---|
| Acinetobacter | $0.0238 \pm 0.0658$ | $0.0042 \pm 0.0103$ | 9/35 | 0.256 |
| Agrobacterium | 0 | 0 | 0/35 | NA |
| Actinomyces | $1.25 \pm 2.45$ | $5.08 \pm 4.86$ | 35/35 | 0.0187 |
| Bacillus | $0.0723 \pm 0.180$ | $0.00393 \pm 0.000778$ | 12/35 | 0.148 |
| Kocuria | 0 | 0 | 2/35 | NA |
| Rhodococcus | $0.00288 \pm 0.052$ | $0.000216 \pm 0.000780$ | 5/35 | 0.0586 |
| Staphylococcus | $0.0377 \pm 0.039$ | $0.219 \pm 0.283$ | 32/35 | 0.038 |
| Kingella | $0.0943 \pm 0.166$ | $0.0702 \pm 0.179$ | 27/35 | 0.832 |
| Micrococcus | $0.00299 \pm 0.0087$ | $0.00194 \pm 0.00468$ | 25/35 | 0.861 |

Notes.
[a] Mean level of the taxa.
[b] Number of subjects with the genus at detectable levels.
[c] Probability of difference in never smoker vs. smoker levels using Welch's $t$-test.

**Table 2  Levels of PAH-selected bacteria on tongue.**

| Genus | Percentage level[a] Never smoker | Percentage level Smoker | Detected[b] | Probability[c] |
|---|---|---|---|---|
| Acinetobacter | 0 | 0 | 0/41 | NA |
| Agrobacterium | 0 | 0 | 0/41 | NA |
| Actinomyces | $2.31 \pm 3.62$ | $5.29 \pm 5.76$ | 41/41 | 0.092 |
| Bacillus | $0.00456 \pm 0.0054$ | $0.00798 \pm 0.0272$ | 7/41 | 0.639 |
| Kocuria | 0 | 0 | 0/41 | NA |
| Rhodococcus | 0 | 0 | 0/41 | NA |
| Kingella | $0.0178 \pm 0.0323$ | $0.0300 \pm 0.0756$ | 26/41 | 0.955 |
| Micrococcus | 0 | 0 | 0/41 | NA |
| Staphylococcus | $0.07738 + 0.0807$ | $0.182 + 0.214$ | 40/41 | 0.078 |

Notes.
[a] Mean level of the taxa.
[b] Number of subjects with the genus at detectable levels.
[c] Probability of difference in never smoker vs. smoker levels using Welch's $t$-test.

microbiome and if they were indeed elevated on oral mucosal surfaces in smokers. Mucosal surface samples were taken from two sites, the gingiva and lateral border of the tongue. These sites were chosen because they are common sites of tobacco-associated OSCC. A total of 37 donors, 15 smokers (average age 53.6, nine male, six female) and 22 nonsmokers (average age 42.1, 12 male, 10 female) were selected. Of the eight genera identified in the earlier assay only *Staphylococcus* and *Actinomyces* were present at higher than 0.01% of oral bacteria genera at the two mucosal sites (Tables 1 and 2). Notably both taxa are found at higher levels on smoker *versus* nonsmoker gingiva, and trend that way on the tongue (Tables 1 and 2). Identification at the genus level does not assure identification at the species or strain level.

## PAH metabolites produced by oral microorganism *in vitro*

Of the candidate microorganisms that showed ability to survive PAH exposure in the *in vitro* test, the yeast *C. albicans* was the most commonly identified in the assay and is well known to be at appreciable level in the oral mucosa of a subset of both smokers and nonsmokers (*Darwazeh, Al-Dwairi & Al-Zwairi, 2010*; *Sheth et al., 2016*). Two separate isolates of *C. albicans* obtained from two different tobacco users were incubated with PAHs as sole carbon source. In this experiment the PAHs tested were phenanthrene and pyrene. These chemicals were chosen because they can be found at relatively high levels in the oral cavity in tobacco users (*Darwazeh, Al-Dwairi & Al-Zwairi, 2010*; *Sheth et al., 2016*) and they are readily digested by many environmental microbes (*Ghosal et al., 2016*; *Hadibarata et al., 2017*; *Hennessee & Li, 2016*; *Hesham et al., 2009*; *Husain, 2008*; *MacGillivray & Shiaris, 1993*; *Mallick, Chakraborty & Dutta, 2011*; *Mbachu, Chukwura & Mbachu, 2016*; *Sowada et al., 2014*; *Sutherland, 1992*). A soil-derived strain, *Mycobacterium rutilum*, capable of metabolizing pyrene or phenanthrene as the sole carbon source, was used as a positive control. Comparative metabolism studies clearly showed that *C. albicans* does not oxidize PAHs to a significant extent while mycobacteria readily do so (Fig. 1). The data indicates that the metabolism of the phenanthrene in the mycobacterium is nearly complete. Only $1.3 \pm 1.6\%$ of the phenanthrene remains after incubation. Pyrene was more difficult for the mycobacterium to digest. Here $29 \pm 6\%$ of the starting concentration remained after two weeks of incubation. The recoveries of the PAHs from the *C. albicans* ($76 \pm 10\%$ and $91 \pm 3\%$) are like those from the un-incubated matrices ($83 \pm 7\%$ and $84 \pm 11\%$). The data suggests that *C. albicans* does not degrade PAH at detectable levels when compared to the mycobacteria.

**Characterization of metabolic products:** The most abundant potential phenanthrene and pyrene metabolites formed in *C. albicans* and *Mycobacterium rutilum* incubations were assumed to have hydroxyl groups (*Ghosal et al., 2016*; *Mallick, Chakraborty & Dutta, 2011*). Therefore, extracts were silylated to enhance the metabolite response in the gas chromatography/mass spectrometry (GC/MS) analysis.

## Phenanthrene

The major metabolites found in the analysis of the positive control *Mycobacterium rutilum* incubation of phenanthrene were 9,10-dihydro-9,10-phenanthrenediol, 9,10-phenanthrenedione, and dihydroxyphenanthrene. A mass spectrum of the trimethylsilyl derivative of the 9,10-dihydrophenanthrene diol generated by the *Mycobacterium* is shown in Fig. 2. Analysis of chromatograms derived from extracts from oral yeast *C. albicans* and phenanthrene co-incubation failed to detect similar, or any, metabolites.

The molecule ion is m/z 356. Characteristic fragment ions include a loss of $CH_3$ (m/z 341) and an ion at m/z 117 ($(CH_3)_3SiCH_2CH_3^+$). We also observed a compound with a fragmentation pattern similar to the diol in Fig. 2 eluting at 9.95 min with a molecular mass two daltons less. The mass spectra suggested that this compound was the 9,10-dihydroxypyrene (Fig. 3).

The molecule ion is observed at m/z 354 and the ions at m/z 339 and m/z 117 were formed in the same way as in the spectra of the derivatized diol in Fig. 2. The 9,10-dihroxypyrene

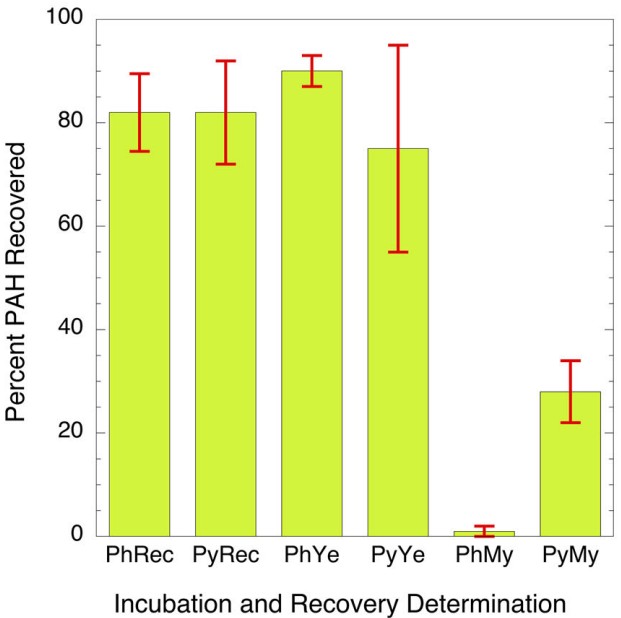

**Figure 1 Bar graph showing percent of parent PAH extracted from mixtures of microorganisms and incubation broth.** PhRec and PyRec, phenanthrene and pyrene recovery from unincubated samples; PhYe and PyYe, phenanthrene and pyrene recovered from incubated yeast samples; PhMy and PyMy, Phenanthrene and Pyrene recovery from incubated *Mycobacterium* samples. The first two columns are the negative controls and represent the recoveries of phenanthrene (PhYe, 83 ± 7%) and pyrene (PyYe, 84 ± 11%) from the incubation matrix. The two bars in the center represent the amount of PAH recovered from *C. albicans* after incubation (PhYe and PyYe). The last two bars represent the amount of each PAH recovered from the *Mycobacteria* incubations (PhMy and PyMy).

was detected in the study of mycobacteria metabolism by GC/MS as a dimethyl derivative (m/z 238) (*Hennessee & Li, 2016*). Figure 3 mass spectra also shows characteristic ions at m/z 264 formed by the loss of $(CH_3)_3SiOH$ as well. In human hepatic liver (*Huang et al., 2014*; *Matsunaga et al., 2009*) and yeast (*Rodriguez, Sobol & Schiestl, 2008*) cells, 9,10-dihydroxyphenanthrene is in a redox equilibrium with 9,10-phenanthrenedione (Fig. 4). Under aerobic conditions, formation of the dione is favored (*Huang et al., 2014*). Therefore, if we find the 9,10-dihydroxyphenanthrene we should observe the dione as well. The molecular mass of the 9,10-phenanthrene dione is 208 daltons. The m/z 208 ion was observed in the mycobacterium sample (Fig. 4A) but not in the *C. albicans* (Fig. 4B) shown below. This observation provides more evidence that *C. albicans* was not oxidizing PAHs.

### Pyrene

Shown are spectra from products produced by positive control *Mycobacterium* digestion of pyrene (Fig. 5). No similar products, nor any other products, were detectable after pyrene incubation with *C. albicans*. In contrast, pyrene formed both mono- and dihydroxy metabolites in the mycobacterium incubations. Two silylated dihydroxy pyrene metabolites with molecule ions at m/z 380 were observed with retention times of 10.29 and 10.50 min, respectively (Fig. 5). The mass spectra of the two compounds were similar (Fig. S1).

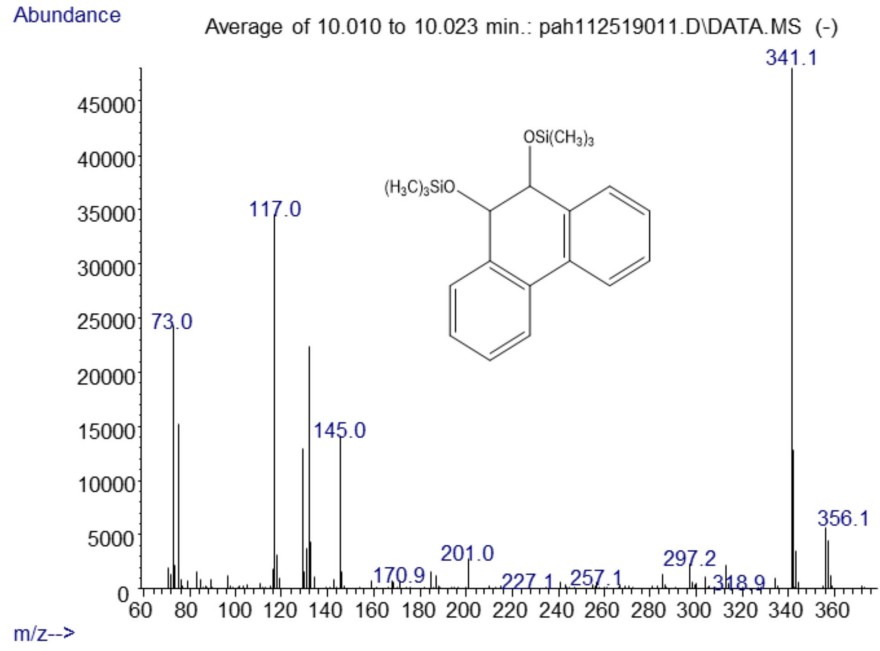

**Figure 2** Electron ionization mass spectrum of compound eluting at 10 min suggested a silylated phenanthrene diol from incubation of *Mycobacterium rutilum* with phenanthrene.

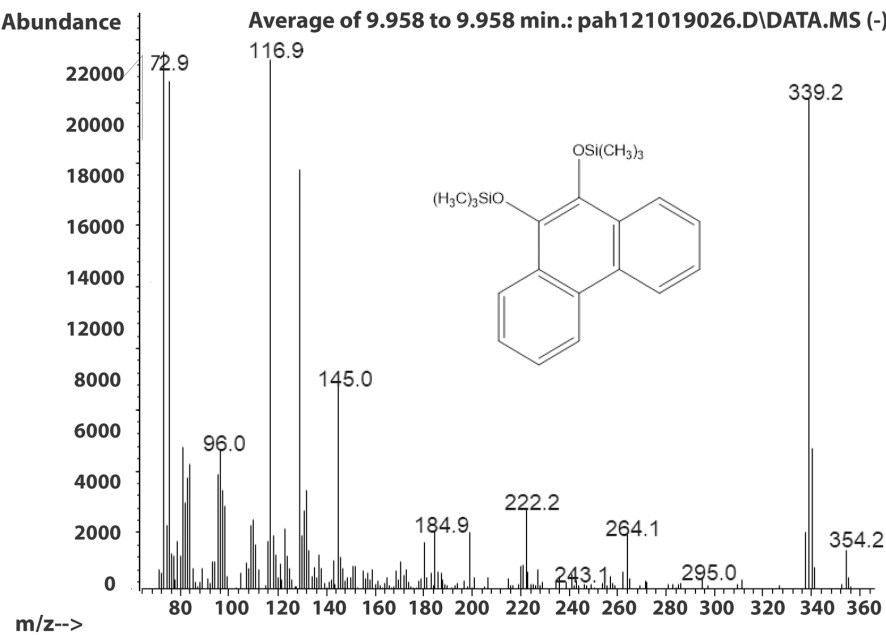

**Figure 3** Electron ionization mass spectrum of compound eluting at 9.95 min suggested to be silylated 9,10-dihydroxyphenanthrene from incubation of *Mycobacterium rutilum* with phenanthrene.

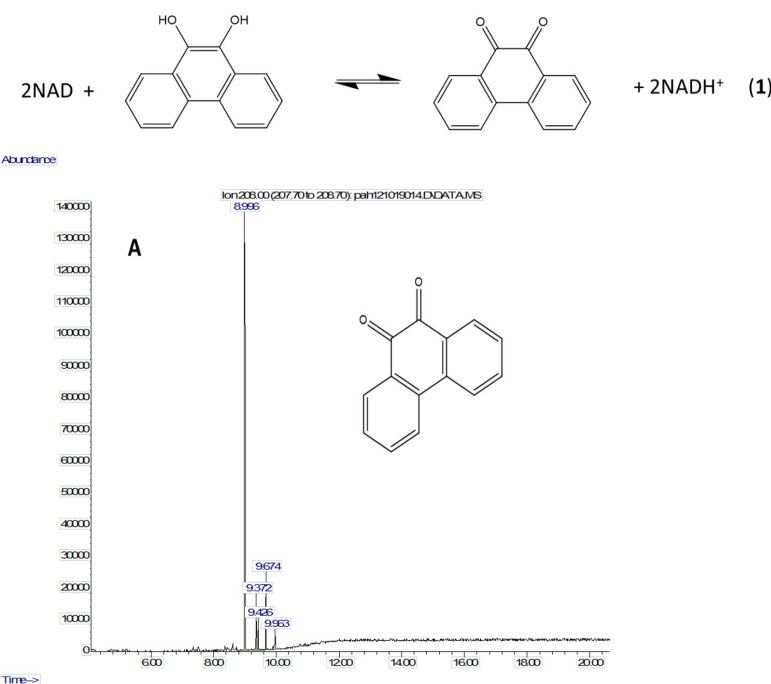

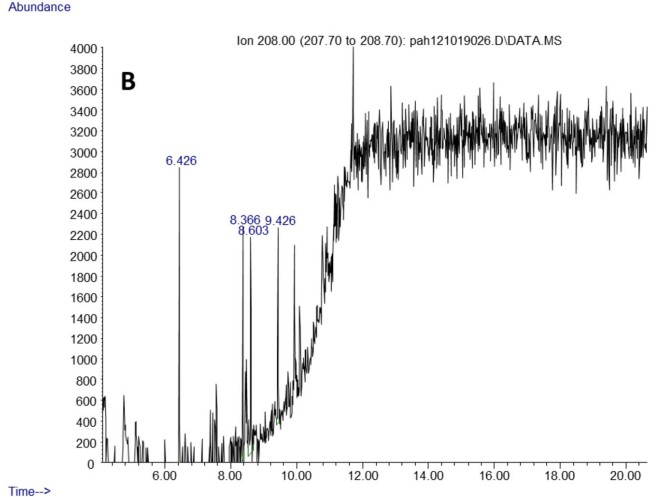

**Figure 4** **Metabolites of phenanthrene produced by *Mycobacterium rutilum* include 9,10-dihydroxyphenanthrene and 9,10-phenanthrenedione.** In aerobic conditions 9,10-dihydroxyphenanthrene and 9,10-phenanthrenedione are in equilibrium. Extracted ion chromatograms of m/z 208 (the molecule ion of 9,10-phenanthrene quinone) derived from the GC/MS analysis of phenanthrene incubated with (A) *Mycobacterium rutilum* and (B) *C. albicans*.

These two compounds were most likely the silylated trans (Rt 10.29) and cis (Rt 10.50) 4,5-dihydroxy-4,5-dihydropyrene. *Heitkamp et al. (1988)* used HPLC, mass spectrometry, and nuclear magnetic resonance (NMR) to determine the structures of the two most
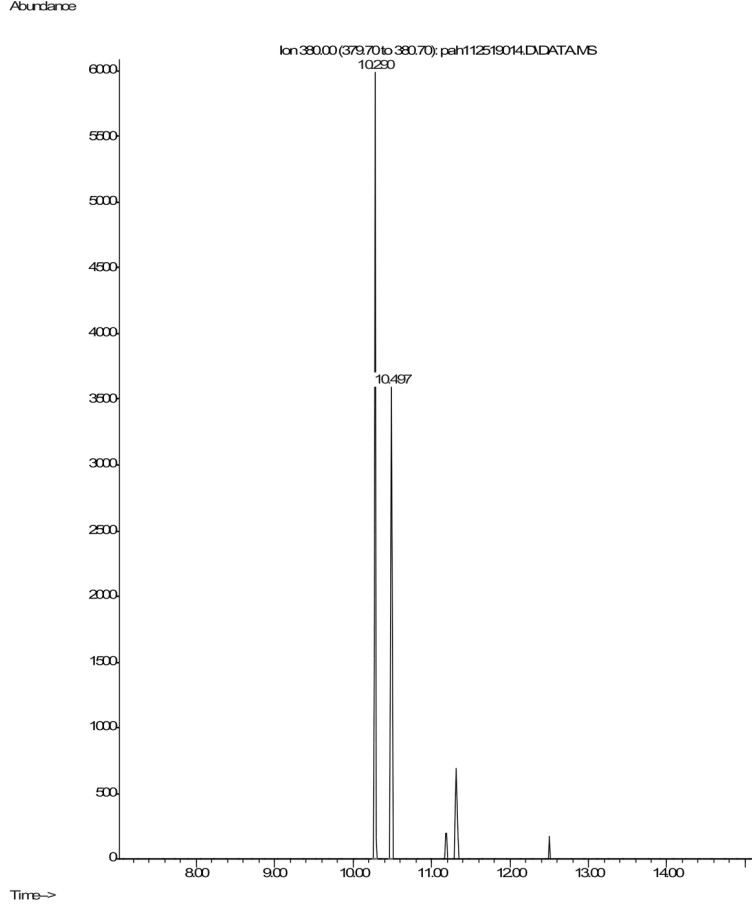

**Figure 5  Extracted ion chromatogram of m/z 380 derived from the GC/MS analysis of pyrene incubated with *Mycobacterium rutilum*.**

abundant *Mycobacterium* pyrene degradation products. The trans isomer was found to be more abundant and have a shorter retention time than the cis isomer.

Two mono-hydroxypyrene metabolites were detected in the mycobacterium incubation as well. The most abundant was determined to be 1-hydroxypyrene (Fig. S2A) through co-chromatography with a standard compound. 1-hydroxypyrene is a human urinary metabolite of pyrene that is frequently monitored to assess PAH exposure in humans (*Flores-Ramirez et al., 2021*; *Jain, 2021*). The most abundant peak in the Fig. S2 mass spectra was the molecule ion at m/z 290. We also observed a compound eluting at Rt 10.84, whose mass spectra was similar to that of the silylated 1-hydroxypyrene (Fig. S2). There are three possible mono-hydroxy isomers. To our knowledge, the 2- and 4-hydroxy isomers have not been observed as metabolites in any previous study of pyrene metabolism. The abundance of the mono hydroxy isomer in Fig. S2 is about 1% of the 1-hydroxypyrene shown in Fig. S2.

### Revised assay to screen for PAH metabolizers

Because there was no evidence that *C. albicans* was capable of digesting pyrene or phenanthrene, we repeated the selection of mucosal oral microbes for growth on PAH using a revised approach to better control for microbes that survive without metabolizing PAH. Incubation after inoculation of a mucosal swab sample into liquid culture was done for 5 weeks with PAH as sole added carbon source and then the bacterial DNA isolated directly without further subculturing on agar plates. All incubations, and not just a subset, were done side by side against a similar assay without the PAH carbon source as a negative control. Donors who were smokers were on average 55.0 years old, and six out of 12 smokers were female. For nonsmokers the average age was 54.2 and five out of eight were female.

Under these conditions, surprisingly, some tongue bacteria had the ability to survive at relatively high numbers over the 5 weeks despite lack of added nutrients. The 16S rRNA amplicon NGS sequencing of the PAH containing and control cultures lacking PAH allowed identification of the taxa that dominate the cultures. *f_Enterobacteriaceae* was identified at high levels in many cultures whether PAH was present or not (Table S1). Importantly, some taxa were observed at high numbers (16 to 95% of the reads) only in the presence of PAH and were not detected in the matched culture without PAH nor in any control culture without PAH (Fig. 6). Six out of 12 tongue biofilm samples fulfilled these criteria of having at least one taxon that showed specific PAH growth. *Micrococcus luteus* was present in samples from three donors, g_*Kingella* in three donors, and g_*Microbacterium* in one donor, all tobacco smokers. Of these, *M. Luteus* and *Kingella* enrichment was statistically significant based on DeSeq2 and/or EdgeR analysis at FDR < 0.1 with Benjamini–Hochberg correction for multiple testing. One nontobacco user out of eight tested showed a single taxon, *Breviabacterium paucivorns,* as enriched after PAH selection and not in the negative control, without PAH. The Fisher Exact test revealed the differences in the two groups does not reach statistical significance at $p < 0.106$ though it trends toward that with an odds ratio of 14 for being positive in the PAH selection test if one smokes *versus* does not smoke. If one were to pool the data from the earlier experiment selecting for PAH surviving microbes with these results, and compare smokers and non-smokers, $N = 40$, there is a difference in the smokers at $p < 0.0005$ , with an odds ratio of 11.4 and an effect size of 1.34.

The dependence on PAH for growth is less clear with *Haemophilus parainfluenzae and Aggregatibacter.* These were enriched specifically in the presence of PAH in two and three, respectively, out of the 10 tobacco user samples but were found in at least one culture grown without PAH from another donor.

## DISCUSSION

*In vitro* selection of bacteria that survive in minimal media in the presence of either a single PAH or a mixture of PAHs is a common assay to select for potential PAH-degrading environmental bacteria (*Juhasz, Stanley & Britz, 2000*; *MacGillivray & Shiaris, 1993*; *Mbachu, Chukwura & Mbachu, 2016*; *Sowada et al., 2014*). Surprisingly, when applied
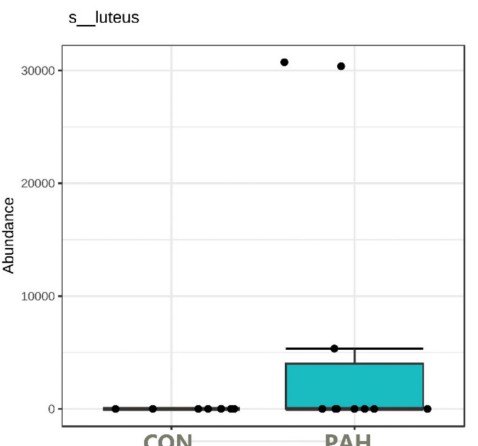
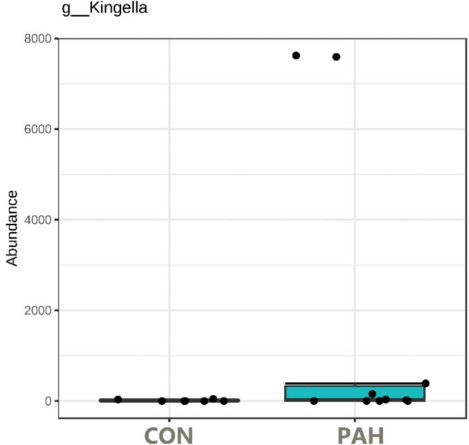

**Figure 6** **Box plot of tongue biofilm taxa that survived 5 weeks of PAH selection and were at zero or minimal levels in culture not exposed to PAH.** Levels based on read counts in 16S rRNA amplicon sequencing assay are shown only for tobacco users. CON cultures lack PAH selection.

to oral isolates, taxa not commonly associated with the oral cavity, such as *Bacillus pumilus, Bacillus subtilis, Rhodococcus, Micrococcus luteus*, and *Agrobacterium tumefaciens*, were identified (*Al-Hebshi et al., 2015*). It is important to note that *Bacillus pumilus, Bacillus subtilis, Acinetobacter*, and *Staphylococcus epiderimidis* have been shown to be present in cigarette tobacco (*Chopyk et al., 2017*; *Rooney et al., 2005*; *Sapkota, Berger & Vogel, 2010*). Overall, this may lead to conjecture that the reason why cigarette smokers had more of these bacteria in the *in vitro* assay is that they are being contaminated on a continuing basis through constant handling of cigarettes.

Bacterial species typically found in soil, *Acinetobacter baumannii, Acinetobacter junii, bacillus subtilis, Bacillus pumilus, Kocuria rhizophila*, or skin such as *M. luteus*, have been shown to be able to metabolize pyrene, naphthalene, and/or phenanthrene in related assays (*Ali et al., 2011*; *Chen et al., 2010*; *Hunter et al., 2005*; *Toledo et al., 2006*). The same is true for some species of *Rhodococcus* but it is not clear how many species of this genus have that property (*Di Gennaro et al., 2014*; *Muangchinda et al., 2017*). With rare exceptions, these taxa are maintained at low levels in the oral cavity. *Acinetobacter* and *Bacillus* genera were detectable at approximately 0.02% of the gingival and lingual biofilms. *Rhodococcus* and *Kingella* genera were easily detectable in a small minority of donors at levels approaching 0.1%. *Staphylococcus epidermidis* in oral samples from two different donors was selected by PAH. Some species of *Staphylococcus*, such as *Staphylococcus epidermidis*, have been shown to be capable of metabolizing PAH (*Moscoso et al., 2012*; *Ye et al., 2014*). *Staphylococcus* as a genus was shown to be enriched 6x to 0.3% of the bacteria genera on gingival mucosa of cigarette smokers, suggesting their resistance to PAH toxicity may select for them *in vivo* (Table 1). Along with *Actinomyces,* both genera are at higher levels on the gingiva of smokers *versus* nonsmokers, and trend that way on the tongue (see Tables 1 and 2). However, the level of the specific species that could metabolize PAH could be lower.

Both PAH selection procedures used in this study revealed smokers have bacteria on their oral mucosal capable of metabolizing PAH, but most taxa are probably at too low a level to cause much damage. While none of the donors had acute periodontal disease, chronic periodontal disease occurs at higher levels among tobacco users in both donor clinics, so it would be difficult to differentiate the possible effects of smoking and periodontal disease on PAH metabolism in this study without a much higher N.

It is of interest that *Kingella* was selected in the PAH metabolizing assay of samples from three smokers highlighting its possible importance. Earlier work has shown that high levels of Kingella while not affected by tobacco usage, were associated with a reduced rate of head and neck squamous cell carcinoma (HNSCC) (*Kurtzman & Robnett, 1997*). Levels of this oral commensal are associated with a host SNP enriched in oral squamous cell carcinoma patients (*Yang et al., 2022*). One could speculate, with very limited evidence, that high levels of Kingella reduce oral squamous cell carcinoma risk, by processing and helping to inactive PAH procarcinogens before they can lead to damage.

This work can be compared to that of *Sowada et al., (2014)* who used a similar approach to study skin microbes. Their 4-week assay of skin microbes with PAH(s) as sole carbon source revealed multiple common core species from the skin of each human subject were able to thrive including skin commensal *M. luteus*. Furthermore, most were capable of completely metabolizing benzo[a]pyrene and about half able to proliferate when grown on phenanthrene. Differences in the assay may contribute to the findings as *Sowada et al. (2014)* and *Sowada et al. (2017)* used benzo[a]pyrene as the sole carbon source in the selection, while in this study a mixture of five PAHs was used. Nevertheless, it seems that PAH metabolizers that thrive when incubated in minimal media with PAH under the condition used, are more readily isolatable from skin than the oral mucosa.

The yeast *C. albicans* was also found to survive the PAH assay and was more frequently selected in samples from smokers than nonsmokers. Members of the *Candida* genus, including *Candida tropicalis, Candida maltosalike*, and *Candida viswanathii,* isolated from environmental sites have been shown to be capable of metabolizing pyrene, phenanthrene, benzo[a]pyrene, napthalene, *etc.* (*Deng et al., 2010*; *MacGillivray & Shiaris, 1993*; *Sutherland, 1992*) though it was not known if *C. albicans* had that ability (see Fig. 6). Because of the high level of *C. albicans* in the mouths of many people, an in-depth analysis of oral *C. albicans* strains' ability to metabolize PAHs was undertaken (*Darwazeh, Al-Dwairi & Al-Zwairi, 2010*; *Sheth et al., 2016*).

Preliminary studies revealed of benzo[a]pyrene, chrysene, fluoranthrene, phenanthrene, and pyrene assayed separately with *C. albicans* isolated from the oral cavity of a tobacco user, none produced obvious products (data not shown). To allow higher resolution in the assay, the focus was put on two model PAHs known to be metabolized by other *Candida* species (*Ali et al., 2011*; *Hadibarata et al., 2017*; *MacGillivray & Shiaris, 1993*; *Muangchinda et al., 2017*; *Wang et al., 2007*). In four repetitions of this experiment minimal loss of starting material occurred when pyrene or phenanthrene PAHs were incubated over 14 days with *C. albicans* isolated from smokers. Any losses of material were similar to that seen with negative controls that lacked microbes but were similarly incubated and isolated. This

contrasts with the *Mycobacterium rutilum*, which metabolized 98% of the phenanthrene and 79% of the pyrenes.

An initial rough scan revealed no obvious metabolites from pyrene or phenanthrene incubated with *C. albicans*. For the positive control, *Mycobacterium rutilum* incubation with phenanthrene, detected products included 9,10- phenanthrene quinone, dihydroxyphenanthrene, and phenanthrene diol (*Hennessee & Li, 2016*). For pyrene, incubation with *Mycobacterium rutilum* produced 4,5-dihydro-4,5-pyrenediol, 1,2-dihydro-1,2-pyrenediol, 2-hydroxypyrene and 4-hydroxypyrene. Unlike the *Mycobacterium rutilum* incubation with phenathrene, little quinone was formed. Analysis done at higher resolving power to look for possible oxidation products of the two PAHs by the oral *C. albicans* revealed none.

The *C. albicans* isolates studied were from the mucosa of tobacco users, assumed to have continual exposure to PAHs, and were subjected to an additional selection for 3 weeks in a mixture of PAHs including these two substrates. Finally, they were given two weeks incubation with either pyrene or phenanthrene to allow detection of the PAH metabolizing activity. The simplest conclusion is that despite their resistance to PAH incubation there is a lack of PAH metabolizing activity in the *C. albicans* strains tested, at least for the model substrates used. It is possible that *C. albicans* survived the initial PAH survival assay by metabolizing components from the other microbes (*Contaldo et al., 2023*; *Kubota et al., 2008*; *Moussa et al., 2021*). It is also possible that under conditions we did not test it can metabolize some PAHs (*Atagana, Haynes & Wallis, 2003*; *Boonchan, Britz & Stanley, 2000*; *Feng et al., 2014*; *Ghosal et al., 2016*; *Suja et al., 2014*).

Using a colorimetric assay to identify anthracene metabolizers based on NADH oxidation detection, it has been reported that several saliva bacteria, especially from tobacco users and including *Streptococcus mutans*, *Lactobacillus fermentum*, *Lactobacillus salivarius* and *Veillonella tobetsuenis*, can degrade that PAH over an 8 day period (*Kubota et al., 2008*; *Moussa et al., 2021*). Three of the four microbes with this property are thought to reside on teeth, the exception being *V. tobetsuensis* which is found on the tongue (*Mashima & Nakazawa, 2013*). *Lactobacillus fermentum* and *Streptoccocus mutans* which can be fairly abundant on teeth were not selected in our assays which examined PAH metabolizers from mucosal surfaces, nor was *V. tobetsuensis* which is a strict anaerobe and would be unlikely to survive the assays described in the current study.

Two variations of an assay for survival of mucosal bacteria with PAH as sole carbon source were done in the current study. The first tested for survival over 3 weeks followed by plating on agar plates with rich broth and counting of colonies. Positive mucosal samples produced many colonies while several samples tested without PAH addition, as controls, produced few. The second assay extended the length of time of the PAH incubation to 5 weeks, avoided colony formation as a read out, had negative controls for each mucosal sample, and used whole population 16S rRNA sequencing for taxa identification and measurement of abundance. The criterium for positivity was that taxa be >10% of reads in the culture post-incubation and, making it more conservative than the first assay, lack of that taxon's appearance in any negative controls without PAH. For that reason, bacteria species that metabolized PAH but were also saprophytic and can survive even at a low level

without PAH would not be counted. In addition, bacteria that metabolize PAH but do not survive long term in the harsh assay conditions would also be negative. The alternative colorimetric assay, for NADH oxidation, which is easier and is likely more sensitive, would likely expand the number of mucosal taxa that are potential PAH metabolizers (*Kubota et al., 2008*; *Moussa et al., 2021*). Due to its usage of a reporter system with the indirect measurement of metabolites, results would need to be verified with another assay. Another alternative, gene-based identification of PAH metabolizers among oral taxa may be another approach in the future when PAH metabolizing enzymes are more clearly understood in nonenvironmental bacteria (*Sakshi & Haritash, 2020*). In the end direct assays to verify PAH metabolic products, such as that done here with *C. albicans*, are needed, and the assays need to be done aerobically and anaerobically. Limitations of this approach are that the assay is cumbersome so that only select PAHs and conditions can be tested, and because *in vitro* analysis may not reflect what occurs *in vivo*.

A fundamental question that remains to be answered is if specific PAH metabolizing microbes increase or decrease the carcinogenesis process brought by tobacco smoking. An understanding of PAH metabolic products from *in vitro* studies is a first step. *In vivo* studies to determine the location and longevity of these newly made carcinogens in the oral environment should more fully address that question.

## CONCLUSIONS

A number of bacteria found on oral mucosal sites are likely PAH metabolizers. Species of the genera *Staphylococcus*, *Actinomyces* and *Kingella* remain as potential candidates to be fairly abundant PAH metabolizers that reside on the oral mucosa with the most support for *Kingella*. Most oral mucosal PAH metabolizers identified in this study were at very low levels. *C. albicans* was a prime candidate to be an abundant PAH metabolizer, which could contribute to oral carcinogenesis, as it is abundant on the mucosa of some smokers, and has taxonomic relatives that can metabolize PAH. However, in direct assays it appeared to largely lack the ability to metabolize PAH model substrates phenanthrene and pyrene under the conditions tested (*Hennessee & Li, 2016*; *Sowada et al., 2017*; *Sowada et al., 2014*). The possibility remains that abundant PAH metabolizing mucosal microbes may be identified by testing for PAH metabolism in the presence of additional substrates or under anaerobic conditions (*Atagana, Haynes & Wallis, 2003*; *Boonchan, Britz & Stanley, 2000*; *Feng et al., 2014*; *Ghosal et al., 2016*; *Suja et al., 2014*).

## ACKNOWLEDGEMENTS

We thank Catherin Rafin, Université du Littoral Côte d'Opal, for advice on PAH metabolizing environmental microbes.

### Funding

This work was supported by the National Institute of Environmental and Health grant R41ES025528 to Arphion Ltd and University of Illinois Chicago. There was no additional external funding received for this study. The funders had no role in study design, data collection and analysis, decision to publish, or preparation of the manuscript.

### Grant Disclosures

The following grant information was disclosed by the authors:
National Institute of Environmental and Health: R41ES025528.

### Competing Interests

The authors declare they have no competing interests.

### Author Contributions

- Lin Tao conceived and designed the experiments, authored or reviewed drafts of the article, and approved the final draft.
- M Paul Chiarelli conceived and designed the experiments, performed the experiments, analyzed the data, prepared figures and/or tables, authored or reviewed drafts of the article, and approved the final draft.
- Sylvia Pavlova performed the experiments, analyzed the data, authored or reviewed drafts of the article, and approved the final draft.
- Antonia Kolokythas performed the experiments, authored or reviewed drafts of the article, and approved the final draft.
- Joel Schwartz conceived and designed the experiments, prepared figures and/or tables, and approved the final draft.
- James DeFrancesco performed the experiments, authored or reviewed drafts of the article, and approved the final draft.
- Benjamin Salameh performed the experiments, prepared figures and/or tables, and approved the final draft.
- Stefan J. Green performed the experiments, authored or reviewed drafts of the article, and approved the final draft.
- Guy Adami conceived and designed the experiments, performed the experiments, analyzed the data, prepared figures and/or tables, authored or reviewed drafts of the article, and approved the final draft.

### Human Ethics

The following information was supplied relating to ethical approvals (*i.e.*, approving body and any reference numbers):

University of Illinois Chicago Institutional Review Board 1.

### Data Availability

The complete taxa abundance table generated from 16S rRNA sequence data from PAH selection is available in the Supplemental File.

## Supplemental Information

Supplemental information for this article can be found online at http://dx.doi.org/10.7717/peerj.16626#supplemental-information.

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
