# Peer review of "Enrichment of polycyclic aromatic hydrocarbon metabolizing microorganisms on the oral mucosa of tobacco users"

_PeerJ, doi:10.7717/peerj.16626_

## Round 0.1 · original submission · Major Revisions

Please address the comments raised and resubmit

Reviewer 1 ·

Basic reporting

The manuscript characterizes PAH metabolizing microorganisms and metabolized products of PAH in the oral mucosa of tobacco users. The article is reasonably organized. It begins with the topic and hypothesis and outlines the methods and conditions under which the microbes were tested. However, there are a few major and minor concerns that need to be addressed before it can be considered for publication.

Experimental design

The manuscript is within the scope of the journal. More clarification is required on the research question and how this research fills the identified knowledge gap.
Major concerns:
Line 79-80: what do the authors mean by “….microbes tolerant of tobacco smoke PAHs”? Do the authors mean that these microbes are able to metabolize PAHs or survive under high PAH conditions, or something else, please clarify.

Line 83: “….gingival mucosa in people”, do the authors mean smokers?

Lines 74-84: can be moved under the methods or results section.

Line 97: The composition of sampled microbial species may be significantly affected by individual patients' periodontal status, making it difficult to attribute changes to tobacco use or oral disease. Tobacco users have a higher risk of developing oral diseases than non-users. No details on inclusion-exclusion criteria have been included in the manuscript.

Demographic characteristics of the study participants should be included.

Although “samples were taken only from sites that appeared normal in appearance in the clinic”, none of the reported exclusion criteria can exclude patients with active periodontal disease. With no measure of clinical parameters, it is impossible to state that patients were in "good oral health definitively." Notably, individual patients' periodontal status demonstrates the oral environment's inflammation and may significantly affect the composition of sampled microbial species. Therefore it is difficult to conclude if the changes observed are attributable to tobacco use or oral disease.
Tobacco users have a much higher proportion of developing gingival, periodontal disease, or any lesions compared to non-users.

Lines 101-109 need more details as to what procedure was used to collect the swabs from the different sites, duration of swabbing, etc (to compare against procedure #2 as mentioned in line 113, swabbed for 30 secs).

Line 160: why did the authors decide to use the MasterPure Gram Positive DNA kit, which would only extract gram-positive bacteria? Needs clarification.

Line 165: needs information on the primers used in order to replicate the methods.

Lines 172: ‘slight modifications’ are not mentioned for replication of methods. Also, it is not clear why the authors used both V1-V3 and V4 amplification.

Validity of the findings

Major concerns:
Many of the bacteria identified as potential PAH metabolizers in the oral mucosa were found at low levels (line 230, “…less than 0.01%). This raises questions about their actual significance in PAH metabolism and whether they can cause significant harm. Given that there was no difference in the PAH metabolizing bacteria in samples between smokers and non-smokers, the authors should consider editing the title of the manuscript. The only result that corroborates with the title is the culture data where there is an insufficient number of samples (6 non-smokers vs. 14 smokers).

Line 205: the authors do not ‘investigate functional changes’, so please consider editing this sentence. Also, the authors cannot definitely say that the PAHs originated from exposure to tobacco smoke, there might be other confounding variables.

Line 206: no sample size calculation was mentioned.

Line 211: Sanger sequencing details were not mentioned in the methods section.

Line 212: were all 14 tobacco smokers specifically cigarette smokers?

Line 222, 228: There is no detail provided for the number of years these subjects were smoking, there are studies showing differences in the oral microbiome with the number of years smoking. Did the authors record data about how long the smokers used tobacco, and what criteria were used for defining long-term tobacco users”?

The discussion reports the significance of the findings and the implications for carcinogenesis in tobacco users. It concludes that evidence for large-scale microbial degradation of tobacco PAHs remains lacking. Limitations such as the relevance of in vitro conditions to the oral mucosa environment in vivo should be discussed. Please consider including potential avenues for further research in this area. The authors need to provide compelling future directions for this work, including the possible associations between differential microbial communities and local or systemic markers of health or disease.

Additional comments

Minor concerns:
Several full forms are missing: eg. Line 102 PBS; NGS (line 116); YPD (line 120).
Lines 203: needs citation.
Lines 248-253: these details need to be included only in the legend of the figure. Line 258: what ‘significance’ test was performed?
Line 271: Please italicize the genus name
Figure 4 Equation 1 needs to be cited in the text.
Line 294 is incomplete
Lines 207: supplemental figures are not attached.
Line 360: Junii should be starting with lower case letter
Throughout the manuscript, please consider using one word, either ‘subjects’ or ‘donors’
Line 411: please fix the citation “15” to the proper format
Line 434: Please italicize bacteria genus and species

Reviewer 2 ·

Basic reporting

BASIC REPORTING
This article adheres to the basic reporting criteria of the PeerJ journal as follows:
1. The article is written in clear, unambiguous, and professional English.
2. The introduction and background sections provide context for the research.
3. The literature is extensively referenced and highly relevant to the study.
4. The article's structure conforms to PeerJ standards and maintains clarity.
5. Figures included in the article are pertinent, of high quality, and accompanied by clear labels and descriptions.
6. Raw data has been provided."

Experimental design

EXPERIMENTAL DESIGN
1. The study represents original primary research within the scope of the journal.
2. The research question is well defined, relevant, and meaningful. It is clearly stated how the research addresses an identified knowledge gap.
3. The investigation is conducted rigorously, maintaining high technical and ethical standards.
4. The methods are described with sufficient detail and information to allow for replication

Validity of the findings

VALIDITY OF THE FINDINGS
1. The article demonstrates the impact and novelty of the research.
2. Meaningful replication is encouraged where the rationale and benefit to the literature are clearly explained.
3. All underlying data have been provided, and they are robust, statistically sound, and well-controlled.
4. The conclusions are well-stated, directly linked to the original research question, and limited to supporting the results."

Additional comments

I Comments and feedback on Abstract.

1. Clarity of Hypothesis and Objectives: The abstract clearly states the hypothesis that oral mucosal microbes in smokers can metabolize polycyclic aromatic hydrocarbons (PAHs) and contribute to carcinogenesis. The objectives are also well-defined: to isolate bacteria and fungi from the oral mucosa of smokers and nonsmokers capable of surviving in minimal media with PAHs as the sole carbon source under aerobic conditions.
2. Methods and Sample Size: The methods employed for microbial isolation are briefly described. However, the abstract could benefit from additional details regarding the microbial isolation process. Additionally, the sample size appears relatively small, with 26 smokers and 16 nonsmokers, which may limit the generalizability of the findings.
3. Results: The results indicate that certain bacteria genera, such as Staphylococcus, Actinomyces, and Kingella, were more abundant among smokers with PAH exposure. Two strains of Candida albicans were isolated from smokers, but they were not found to metabolize PAHs in the tested conditions. This information is presented clearly.
4. Discussion and Interpretation: The abstract effectively discusses the implications of the findings. It highlights the absence of evidence for large-scale microbial degradation of tobacco PAHs on the oral mucosa under aerobic conditions. It also notes the presence of non-abundant PAH metabolizers.
Conclusion: The abstract concludes by summarizing the key findings. However, it might benefit from a more explicit statement about the overall implications of the study's findings for the field of dental research and oral health.
Overall, the abstract effectively presents the research hypothesis, methods, results, and their implications.

II Comments and feedback on the Introduction

1. Background and Context: The introduction provides a comprehensive background on the effects of tobacco smoke on the oral cavity, particularly focusing on periodontal health, changes in oral microbiota, and the potential link to oral squamous cell carcinoma (OSCC). This context helps readers understand the significance of the study.
2. Citations and References: The introduction effectively incorporates relevant references to support the claims and context. However, it could benefit from more recent references to maintain the relevance of the cited studies.
3. Research Gap Identification: The introduction identifies a research gap by highlighting that while environmental microorganisms capable of metabolizing polycyclic aromatic hydrocarbons (PAHs) are known to exist, there is a need to investigate whether such microbes are present on the oral mucosa, particularly in smokers. This is a clear and logical rationale for the study.
4. Methodology and Approach: The introduction outlines the methodology used in the study, mentioning the selection of microorganisms from petroleum waste sites known to survive in minimal media with PAHs as the sole carbon source. This method is relevant to the research question and provides insight into the study's design.
5. Research Objectives: The introduction effectively conveys the primary research objectives, which are to investigate whether smokers harbor PAH-tolerant microbes on their oral mucosa and to identify specific taxa, such as Micrococcus luteus and Kingella, that might be PAH metabolizers. Additionally, the mention of examining C. albicans' ability to metabolize model PAHs adds clarity to the research goals.
6. Clear Hypotheses: While the introduction presents clear hypotheses about the presence of PAH-tolerant microbes in smokers and the potential role of specific taxa, it could be more explicit in stating the expected outcomes of the study.
7. Language and Clarity: The introduction is generally well-written and clear, but it could benefit from minor grammatical and syntactical adjustments for improved readability.
In summary, the introduction effectively sets the stage for the study by providing relevant background information, identifying research gaps, outlining the methodology, stating clear research objectives, and presenting hypotheses. Incorporating more recent references and making minor language improvements would enhance the overall quality of the introduction.

III Comments and feedback on the Materials and methods

1. Study Subjects and Ethical Considerations: The section provides clear information about the study subjects, their sources, and ethical considerations, including informed consent and approval by Institutional Review Boards . This demonstrates compliance with ethical guidelines and human subject protections.
2. Sample Collection: The section details the collection of mucosal samples from various sites in the oral cavity. The inclusion of sites such as the tongue, buccal, attached gingiva, and oral pharynx adds diversity to the sample sources, providing a comprehensive assessment.
3. PAH Selection Procedure: The procedure for selecting microorganisms with the ability to tolerate and metabolize PAHs is explained clearly. The use of Benzo[a]pyrene, chrysene, fluoranthrene, naphthalene, phenanthrene, and pyrene as PAHs and their incubation conditions are well-defined.
4. Use of Controls: The inclusion of controls, such as BHB medium without PAH and BHB medium with microbial cells without PAH, is essential for valid experimentation. These controls help establish a baseline for the effects of PAH exposure.
5. Identification of Metabolites: The section explains the process of identifying pyrene and phenanthrene metabolites using Gas Chromatography-Mass Spectrometry (GC/MS). The method of derivatization with BSTFA is described clearly, and the inclusion of deuterated standards is appropriate for quantification.
6. DNA Extraction: The section briefly mentions the method used for genomic DNA extraction from swabs. However, more details about the DNA extraction procedure, including any modifications or specific kits used, would provide clarity.
7. Characterization of Microbial Community: The description of microbial community structure analysis through high-throughput sequencing of 16S rRNA genes is adequate. The use of different primer sets for V1-V3 and V4 regions is noted, which can provide insights into bacterial diversity.
8. Bioinformatics Analysis: The description of bioinformatics analysis using QIIME, chimera removal, and OTU clustering is informative. The explanation of taxonomic assignment and the assignment of taxa from phylum to species is beneficial.
9. Statistical Analysis: The statistical analysis section outlines the use of Welch's t-test to identify differentially abundant taxa and mentions significance levels. However, more details about the criteria used for significance and correction for multiple testing (if applied) would enhance the clarity of this section.
10. Sample Size and Replicates: While the methods are generally well-described, there is no mention of the sample size or replicates used in the different experiments. Please Provide this information. Additionally, as per the “abstract” where the sample size is mentioned the sample size appears relatively small, with 26 smokers and 16 nonsmokers, which may limit the generalizability of the findings. Please comment and make necessary changes in the manuscript.

Overall, the "Materials & Methods" section provides a clear and comprehensive description of the experimental procedures and analysis techniques used in the study. Clarifications regarding DNA extraction details and information on sample size and replicates would further improve the section.

IV Comments and feedback on the presented Results:

1. Sample Size and Group Comparison:
• The study involved a total of 37 subjects, with 15 smokers and 22 nonsmokers which is contradictory to the sample size mentioned in the abstract . While the number of subjects seems reasonable here , it would be helpful to provide more details about the demographics and smoking habits of the participants to better understand the study population.
2. Survival of Oral Mucosal Bacteria:
• The study found that PAH selection allowed the survival of specific bacteria in the oral mucosa, and smokers had higher numbers of these PAH-resistant microbes compared to nonsmokers. This finding is interesting and relevant to the study's objectives.
• It's essential to report the statistical significance of these differences, which is briefly mentioned (Fisher's Exact Test, p < 0.018). Providing more detailed statistical analyses and effect sizes would strengthen the results.
3. Identification of Microbes:
• The article mentions several bacteria that survived PAH exposure, including Acinetobacter junii, Acinetobacter baumannii, Actinomyces, Bacillus species, Kocuria rhizophila, Rhodococcus, and Staphylococcus. This information is valuable for understanding the microbial composition.
• The presence of Candida albicans (C. albicans) is noted, and it is essential to highlight that C. albicans is a yeast and not a bacterium. The use of 28S rRNA gene sequencing for C. albicans identification is appropriate.
4. Differences in Oral Mucosal Bacteria:
• The article mentions the presence of Staphylococcus and Actinomyces as the dominant genera on the oral mucosal surfaces. However, it would be helpful to provide more details about the relative abundance of these genera in smokers and nonsmokers.
5. PAH Metabolites Produced by Oral Microorganisms:
• The study investigated the ability of C. albicans to metabolize phenanthrene and pyrene, both of which are commonly found in the oral cavity of tobacco users. The results indicate that C. albicans did not significantly metabolize these PAHs, while Mycobacterium rutilum did.
• The presentation of the results, including the bar graph (Figure 1), effectively communicates the differences in PAH metabolism between C. albicans and Mycobacterium rutilum.
6. Characterization of Metabolic Products:
• The article describes the metabolites formed during the incubation of pyrene and phenanthrene. It is good practice to include mass spectra or chromatograms to support the identification of these metabolites.
7. Revised Assay to Screen for PAH Metabolizers:
• The revised approach to screen for PAH metabolizers is well-described. However, it is essential to provide more information about the specific taxa that were found to be enriched in the presence of PAH, particularly in smokers.
8. Statistical Analysis:
• The article mentions the use of statistical analyses such as DeSeq2 and EdgeR. It would be helpful to include the specific statistical tests used, along with p-values and false discovery rate (FDR) values, to assess the significance of taxonomic enrichment.
Overall, the results section provides valuable insights into the survival of oral mucosal bacteria in the presence of PAHs and their potential role in PAH metabolism. However, providing more detailed statistical analyses and additional context for the taxonomic findings would enhance the clarity and scientific rigor of the study.


V Comments and feedback on the Discussion:

1. Comparative Analysis with Environmental Studies:
• The comparison of the study's findings with environmental studies investigating PAH-degrading bacteria is informative. It highlights the surprising presence of certain bacteria, typically associated with soil or skin, in the oral cavity. The hypothesis that these bacteria might be introduced into the oral cavity through cigarette handling is intriguing and warrants further investigation and needs be mentioned in the manuscript .
Presence of Soil and Skin Bacteria in Oral Cavity:
• The discussion of specific bacterial taxa such as Bacillus species, Rhodococcus, and Agrobacterium tumefaciens being identified in the oral cavity is well-supported by the results. However, it would be beneficial to explain why these bacteria are not commonly found in the oral cavity and their potential sources.
2. Role of Staphylococcus and Actinomyces:
• The discussion regarding Staphylococcus and Actinomyces and their potential resistance to PAH toxicity is relevant. However, to strengthen this point, it would be helpful to discuss the implications of their presence in the context of oral health or disease, if known.
3. Candida albicans Survival and Metabolism:
• The detailed investigation of C. albicans' survival and metabolism of PAHs is well-documented. The results suggest that C. albicans strains isolated from smokers did not significantly metabolize the tested PAHs. This is a valuable finding, and the discussion effectively presents the evidence.
• It's essential to acknowledge the limitations of the study, such as the use of model PAH substrates and the possibility that C. albicans may metabolize other PAHs or components.
4. Comparison with Other Studies:
• The discussion compares the study's findings with similar research on PAH metabolism by oral microbes. Mentioning the use of a colorimetric assay to identify anthracene metabolizers by saliva bacteria adds context to the broader literature.
5. Future Directions and Alternative Assays:
• The discussion briefly touches upon potential future directions, such as the use of alternative assays and gene-based identification of PAH metabolizers among oral taxa. Expanding on these possibilities and their significance would be beneficial.
6. Clarity and Organization:
• The discussion is well-organized and logically structured, making it easy to follow the flow of ideas. However, some paragraphs are quite dense with information, and breaking them into smaller sections could improve readability.
.
Overall, the discussion effectively summarizes the study's findings, places them in context with related research, and provides insights into the potential implications of the results. Addressing the points mentioned above and providing more context regarding the clinical relevance of these findings would further enhance the discussion.

VI Comments and feedback on the Conclusion:
1. Summary of Findings:
• The conclusion effectively summarizes the main findings of the study, which include the identification of potential PAH metabolizers on oral mucosal sites. It highlights the genera Staphylococcus, Actinomyces, and Kingella as potential candidates for PAH metabolism.
2. Supporting Evidence:
• The conclusion appropriately mentions the low levels of most oral mucosal PAH metabolizers identified in the study, emphasizing the challenges in detecting them. This reinforces the need for more sensitive assays or alternative conditions to identify these microbes.
3. Role of C. albicans:
• The conclusion addresses the significant presence of C. albicans on the mucosa of some smokers and its potential role in PAH metabolism. It acknowledges that, under the tested conditions, C. albicans appeared to lack the ability to metabolize PAH model substrates, which is a crucial finding.
4. Future Research Directions:
• The conclusion appropriately suggests potential future research directions, such as exploring PAH metabolism in the presence of additional substrates or under anaerobic conditions. This reflects a scientific approach to further understanding the role of oral microbes in PAH metabolism.
5. Clarity and Conciseness:
• The conclusion is clear and concise, summarizing the findings without unnecessary elaboration.
Overall, the conclusion effectively captures the main takeaways from the study and appropriately addresses the potential significance of the findings. It also highlights the need for further research to explore PAH metabolism under different conditions. This conclusion aligns well with the study's objectives and results.

---

## Round 0.2 · accepted · Accept

Congratulations to the authors.

Reviewer 1 ·

Basic reporting

no comment

Experimental design

no comment

Validity of the findings

no comment

Reviewer 2 ·

Basic reporting

NO COMMENTS

Experimental design

NO COMMENTS

Validity of the findings

NO COMMENTS

Additional comments

Based on the revisions made by the authors, I am satisfied with the current state of the manuscript. The authors have demonstrated commitment to improving the quality of their work.

Therefore, I would like to recommend the manuscript for publication